 **eLIFE**

# Rapid learning in visual cortical networks

Ye Wang, Valentin Dragoi*

Department of Neurobiology and Anatomy, The University of Texas Medical School at Houston, Houston, United States

**Abstract** Although changes in brain activity during learning have been extensively examined at the single neuron level, the coding strategies employed by cell populations remain mysterious. We examined cell populations in macaque area V4 during a rapid form of perceptual learning that emerges within tens of minutes. Multiple single units and LFP responses were recorded as monkeys improved their performance in an image discrimination task. We show that the increase in behavioral performance during learning is predicted by a tight coordination of spike timing with local population activity. More spike-LFP theta synchronization is correlated with higher learning performance, while high-frequency synchronization is unrelated with changes in performance, but these changes were absent once learning had stabilized and stimuli became familiar, or in the absence of learning. These findings reveal a novel mechanism of plasticity in visual cortex by which elevated low-frequency synchronization between individual neurons and local population activity accompanies the improvement in performance during learning.

## Introduction

Although perceptual learning has been a phenomenon studied for many decades, the neuronal mechanisms underlying the improvement in sensory discrimination after practice remain mysterious. Classical theories of perceptual learning have proposed that the improvement in performance during learning relies on a fine retuning and an overrepresentation of the neurons involved in the processing of the trained stimuli (*Sagi and Tanne, 1994*; *Karni and Bertini, 1997*). However, experiments in sensory cortex failed to find an expansion in the cortical representation of the trained stimulus (*Schoups et al., 2001*; *Ghose et al., 2002*) and reported relatively modest changes in neuronal responses and sensitivity after learning (*Schoups et al., 2001*; *Ghose et al., 2002*; *Sigala and Logothetis, 2002*).

An alternative possibility is that learning causes pronounced changes in the way in which information is encoded in population activity, but weaker changes at the single-cell level. Examining the changes in population activity related to learning is motivated by the fact that sensory information is actually encoded in a distributed manner across populations of neurons. Thus, behavioral performance in visual, auditory, or motor tasks (*Sparks et al., 1976*; *Georgopoulos et al., 1986*; *Lee et al., 1988*) is known to be much more accurate than would be predicted from the responses of single neurons (*Paradiso, 1988*). Furthermore, theoretical studies have demonstrated that coding strategies based on the responses of a population of neurons encode more information than coding strategies based on single-cell responses (*Zohary et al., 1994*; *Abbott and Dayan, 1999*; *Pouget et al., 2000*; *Sompolinsky et al., 2001*). Unfortunately, despite the clear importance of analyzing population activity, whether and how networks of cells exhibit changes in population activity to influence the accuracy of behavioral responses during learning has rarely been investigated experimentally.

The major limitation that prevented our understanding of how the information encoded by populations of neurons changes during the time course of learning has been the inability to record from the same population of neurons for extended periods of time, while animals improve behavioral performance during learning. To overcome this limitation, we examine here the relationship between

*For correspondence: Valentin.Dragoi@uth.tmc.edu

Competing interests: The authors declare that no competing interests exist.

**eLife digest** Throughout life, we learn and become better at many skills through repeated practice. However, how the brain cells enable us to adapt to changes in the environment and improve cognitive performance is poorly understood.

The activity of a neuron can be recorded as a 'spike' of electrical activity. In the nervous system, neurons work together in networks. If a group of neurons fire in a synchronized manner, waves of activity may be recorded from that brain region. One important issue in neuroscience is whether the spikes of individual neurons are synchronized with the local network activity. Indeed, it is generally believed that it is functionally important for individual cells to synchronize their responses to the waves of population activity.

The vast majority of studies aimed at understanding the behavior of neurons during learning have only recorded the activity of single neurons. This activity does not change much during learning, which suggests that learning may instead be encoded by the combined activity of a group of neurons. However, it is difficult to examine the same population of neurons as an animal practices and improves a skill. This is because the learning process typically takes longer than the length of time for which a single cell can be held in a stable condition and recorded from.

To overcome these limitations, Wang and Dragoi briefly flashed images at monkeys and trained them to report when the images have been rotated. Monkeys learn to do this within a single-training session, which allows the responses of the same group of neurons—found in a part of the brain called the mid-level visual cortex—to be recorded throughout the learning process.

Wang and Dragoi found that the improvement in behavioral performance during learning was accompanied by a tight synchronization between the spikes produced by individual neurons and the activity of groups of cells within a specific low-frequency band. This low-frequency activity had previously been linked to changes in the strength of functional connections between neurons in the hippocampus, which may be important for learning. The more synchronized this neural activity was, the better the monkeys were at the task. However, changes to the synchronization of spiking responses to local population activity in the higher frequency bands were unrelated to changes in performance. The changes to the level of synchronization were abolished once learning had stabilized and stimuli had become familiar.

Although Wang and Dragoi have found that the mid-level visual cortex neurons fire in a more synchronized way throughout learning, it remains to be confirmed whether these changes in synchronization are causally related to learning. Future studies could test whether this is the case by electrically or optically stimulating neurons so that their activity synchronizes with the local population activity, and investigating whether this manipulation improves learning ability.

learning and population activity by devising a task in which monkeys rapidly learn (within one session) to discriminate between consecutive, briefly flashed images slightly rotated with respect to each other. This task offers the advantage that the same population of neurons can be examined during the entire time course of learning, thus greatly reducing the sampling bias characterizing the day-by-day acute recordings of neuronal activity.

To examine the neural network correlates of rapid learning, we focused on the population response in mid-level visual cortex (area V4, *Desimone and Schein, 1987*; *Hegde and Van Essen, 2005*). Among all sensory cortical areas, the visual cortex is the best understood in terms of receptive field properties and circuitry (*Hubel and Wiesel, 1969*), thus, it provides a unique opportunity for investigating the impact of perceptual learning on neuronal responses. Importantly, area V4 sends inputs to downstream areas involved in perceptual decisions, and individual neuron responses in extrastriate cortex are more strongly correlated to behavior than those in V1 (*Nienborg and Cumming, 2006*; *Gu et al., 2014*). In addition, lesion studies have suggested that area V4 plays a key role in perceptual learning (*Schiller and Lee, 1991*).

## Results

We simultaneously recorded single units and local field potentials (LFP) using multiple electrodes in mid-level visual cortex (area V4) of two macaque monkeys. Monkeys were trained to perform an image

discrimination task (*Figure 1A*) in which a target image presented for 300 ms was compared, after a 800–1200-ms delay, with a 300-ms test image (either identical to the target or rotated by 3°, 5°, 10°, or 20°; the monkeys were required to complete at least 800 trials in one session; all test orientations were randomly interleaved). After monkeys were able to accurately (>85%) perform the task with a set of 10 prototype stimuli (to which they were exposed for many weeks of practice), we introduced novel stimuli at the beginning of each session. Behavioral and electrophysiological data were analyzed in parallel after dividing trials into 96-trial blocks in order to detect changes in performance due to learning (a complete session lasted about 90 min).

*Figure 1B* shows that, as expected, behavioral discrimination threshold gradually decreases during the time course of learning (monkey M1: n = 31 sessions, p = 1.42 $10^{-5}$; monkey M2: n = 17 sessions, p = 0.028, ANOVA test; each of blocks 2–4 is characterized by a lower threshold than block 1, p < 0.05, post hoc multiple comparisons). Interestingly, the largest improvement in behavioral performance occurred between blocks 1 and 2, followed by performance saturation after block 2 (on average, the image orientation discrimination threshold decreased from 18.9° to 4.5°, that is, a 76.2% decrease at the end of the session).

We first examined whether learning induces changes in neuronal responses at individual V4 sites. *Figure 1C,D* shows two examples of spike waveforms and inter-spike interval distribution that demonstrate that our recordings were stable in time. *Figure 1E,F* shows single unit and average LFP traces of example recording channels. We measured the block-by-block changes in mean firing rate, neuronal discrimination performance (d′, defined as the capacity to discriminate between nearby test image orientations), and mean LFP power across blocks of trials. Remarkably, learning did not influence individual neuron responses and LFPs (during the test presentation averaged across all stimulus orientations) in a significant manner. Indeed, *Figure 1G* shows that the population mean firing rate and d′ throughout the test stimulus presentation (n = 105 sites) did not differ during the time course of learning (balanced one-way parametric ANOVA test, p > 0.9 for mean firing rate; p > 0.1 for d′). For individual cells, we found that 15.4% cells increased their firing rates, whereas 17.3% showed a decrease (p < 0.05, ANOVA test, post hoc comparing blocks 1 and 4). Additionally, we compared LFP power changes relative to block 1 but did not find significant differences across blocks of trials in any frequency band (*Figure 1H*, ANOVA and Kruskal–Wallis tests, p > 0.15; we also examined the event-related potentials, or ERPs, for the target and test stimuli, but they did not change across blocks; target: p = 0.66, and test: p = 0.59, ANOVA tests).

Next, we directly tested our hypothesis that the improvement in behavioral performance during learning is accompanied by synchronous firing of neurons with their neighbors. We thus examined the timing relationship between the spikes of single neurons and the ongoing LFP oscillation by quantifying the spike-field coherence (SFC). First, we computed the spike-triggered average (STA, triggered from same number of subsampled spikes in order to avoid bias) by averaging the LFP signal within a window centered ±150 ms on each elicited spike in each block. Second, we computed SFC by dividing the power spectrum of the STA to the average of all power spectra of the LFP segments used to obtain the STA. SFC varies as a function of frequency and yields values between 0 and 1. The larger the SFC, the more accurately the spikes follow a particular phase of this frequency. We calculated the SFC separately for each block (for each single-unit-LFP pair, n = 625). *Figure 2A,B* shows two examples of cross-channel SFC early in the session (block 1) and during learning (blocks 2–4). Clearly, rapid learning is associated with an increase in low-frequency SFC (particularly in the theta band, 4–8 Hz) during the intervals when the two stimuli are presented (0–300 ms and 1300–1600 ms. In contrast, SFC at higher frequency bands (alpha, beta, and gamma bands) was either unchanged or slightly decreased during the time course of learning. These results were confirmed for the population of spike-LFP pairs (*Figure 2C*)—learning was associated with an increase in theta SFC during the intervals when the two stimuli are presented (p < 0.05, Wilcoxon signed-rank test, by comparing theta SFC in blocks 2–4 vs block 1 for time intervals 150–270 ms and 1430–1550 ms; SFC was calculated within a 300-ms window sliding every 10 ms), whereas coherence in the high-frequency bands did not change across blocks of learning (p > 0.05).

Previous studies have reported synchrony modulation mainly during the sustained neuronal response (*Gieselmann and Thiele, 2008*). Therefore, we focused our SFC analysis between 150 and 350 ms after stimulus onset since the transient spike responses, which are highly synchronized to stimulus onset, might possibly induce artifacts related to event-related synchrony. In addition, since neurons have different strengths of their stimulus onset transients, and the SFC measure is sensitive to spike counts,

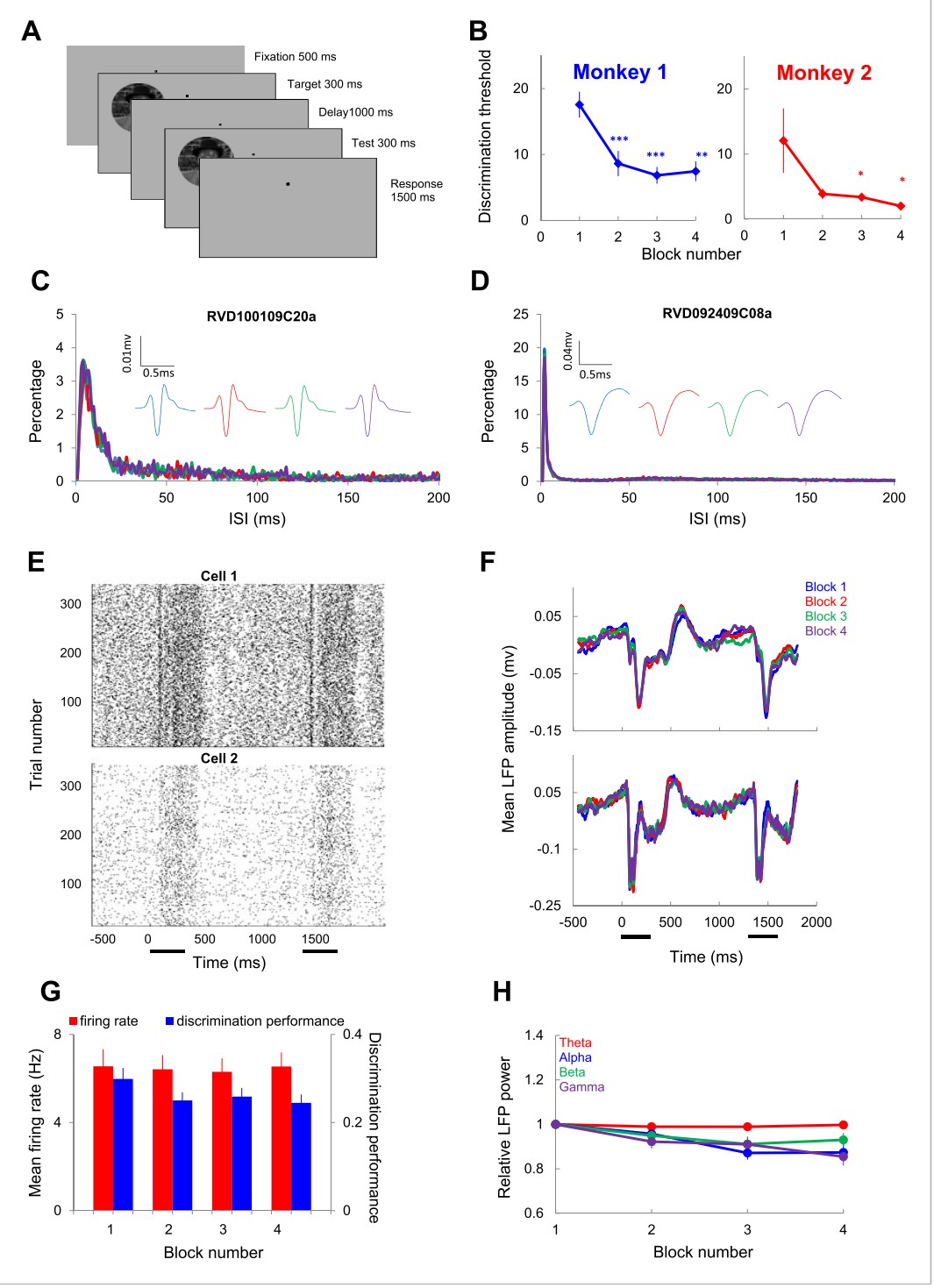

**Figure 1**. Individual neuron and LFP responses in area V4 during rapid learning. (**A**) Schematic description of the rapid learning protocol. After 500 ms of fixation, a target image was presented for 300 ms, followed, after a 1000-ms blank, by a 300-ms test image (monkeys maintained fixation within a 1 × 1 deg window). Monkeys were required to hold a lever for 1500 ms if the test image was rotated relative to the target, and release it within 500 ms if the target and test were identical. (**B**) Changes in mean behavioral discrimination threshold binned in 96-trial blocks for monkeys 1 and 2. Error bars represent s.e.m. (**C**, **D**) Two example inter-spike interval (ISI) distributions for each block in two single units. Insets show average spike waveforms in each block. (**E**) Raster plots represent the spike timing of two V4 neurons recorded throughout the time course of learning. (**F**) Examples of local field potential (LFP)
*Figure 1. continued on next page*

*Figure 1. Continued*

responses at two recording sites. Each curve represents the average response across all trials and test orientations within the same block. The horizontal bars mark the 300-ms time windows during which the target and test stimuli were presented. (**G**) Mean single unit firing rates and discrimination performance (*d'*) across blocks of learning. The error bars represent SEM. (**H**) Mean relative LFP power in the theta, alpha, beta, and gamma bands in different blocks. The LFP power of individual channel in each block across all test orientation was normalized by the mean in the first block; the mean population LFP power was calculated by averaging across all channels. LFP power in different frequency bands was scaled differently since recordings from different sites depended on electrode impedance.

The following figure supplement is available for figure 1:

**Figure supplement 1**. Control experiment—Monkey 2 was passively exposed for 10 sessions to novel natural scenes (similar to those in *Figure 1A*) while the animal performed a color detection task (red–green task) in the contralateral hemifield.

focusing the analysis on the period immediately following stimulus presentation might contaminate our measure of spike-LFP synchronization. We thus computed the mean SFC values (across all pairs and sessions) and found that whereas SFC in the higher frequency bands showed no significant change across blocks (Kruskal–Wallis, $p > 0.2$), theta-band SFC was significantly increased during the time course of learning (*Figure 3A*; $n = 625$ pairs, comparing blocks 2–4 with block 1, Kruskal–Wallis, $p < 10^{-7}$). *Figure 3B* shows the distribution of SFC values for all the pairs in our population and reveals that learning is accompanied by a pronounced increase in theta SFC in blocks 2–4 relative to block 1 ($p < 0.05$, t-test). Furthermore, post hoc analysis shows a significant SFC increase in each of blocks 2, 3, and 4 relative to block 1 ($p < 0.05$; the median theta SFC was increased by 34%). The increase in theta SFC during learning was statistically significant in both animals (*Figure 3C*, $p < 0.001$, Kruskal–Wallis test, post hoc analysis).

One potential concern is that the sensitivity to SFC may differ for different frequency ranges irrespective of the changes observed during learning. For instance, while the theta power is constant across blocks, the power in other bands declines somewhat (*Figure 1H*). In addition, the sensitivity of the coherence results may be related to the number of action potentials falling within a period at each frequency. However, this is not an issue in our analysis. The STAs used in the calculation of SFC is measured by summing all LFP segments and then dividing by the number of spikes. Even though the power spectrum of the STA depends on the power spectrum of the LFP signal (decreasing LFP amplitude decreases the STA power despite the absence of spike-LFP synchronization), SFC is obtained by normalizing the power spectrum of the STA by the average of all power spectra of all LFP segments that were averaged to obtain the STA. This normalization ensures that SFC is independent of the spiking rates and the power spectrum of the LFP. Thus, the small block-by-block changes in LFP power in the alpha, beta, and gamma bands (*Figure 1H*) are unlikely to alter the sensitivity of our SFC measure during learning in the corresponding frequency band.

Another concern is that the changes in theta SFC during learning may simply reflect an ERP with a predominant theta component. Thus, we shuffled the trials and recomputed SFC (in each frequency band)—if the change in theta SFC across blocks simply reflects the stimulus effect, we would expect to find a prominent increase in spike-LFP synchronization even when trials were shuffled. However, as shown in *Figure 3A*, the increase in theta SFC is unrelated to stimulus presentation (although only theta SFC is shown in *Figure 3A*, we did not find statistically significant block-by-block changes in SFC in any frequency band, $p > 0.05$, ANOVA test). We also separated our spike-LFP coherence analysis for the pairs originating from different electrodes and pairs originating from the same or different electrodes. However, although the increase in theta SFC during learning was slightly larger when the spike-LFP pairs were taken from different electrodes ($n = 523$ pairs), the effects were highly significant in both cases (*Figure 3D*, Kruskal–Wallis test). Since LFPs are composed of extracellular voltage fluctuations including local excitatory and inhibitory intracortical inputs (*Leopold and Logothetis, 2003*) originating from recording sites within 2 mm or less (*Katzner et al., 2009*), the modulation in SFC during learning was expected to be more pronounced when the recording sites are close to each other. Although we divided our SFC pairs into near (<2 mm) and far (>2 mm) groups (345 'near' and 280 'far'

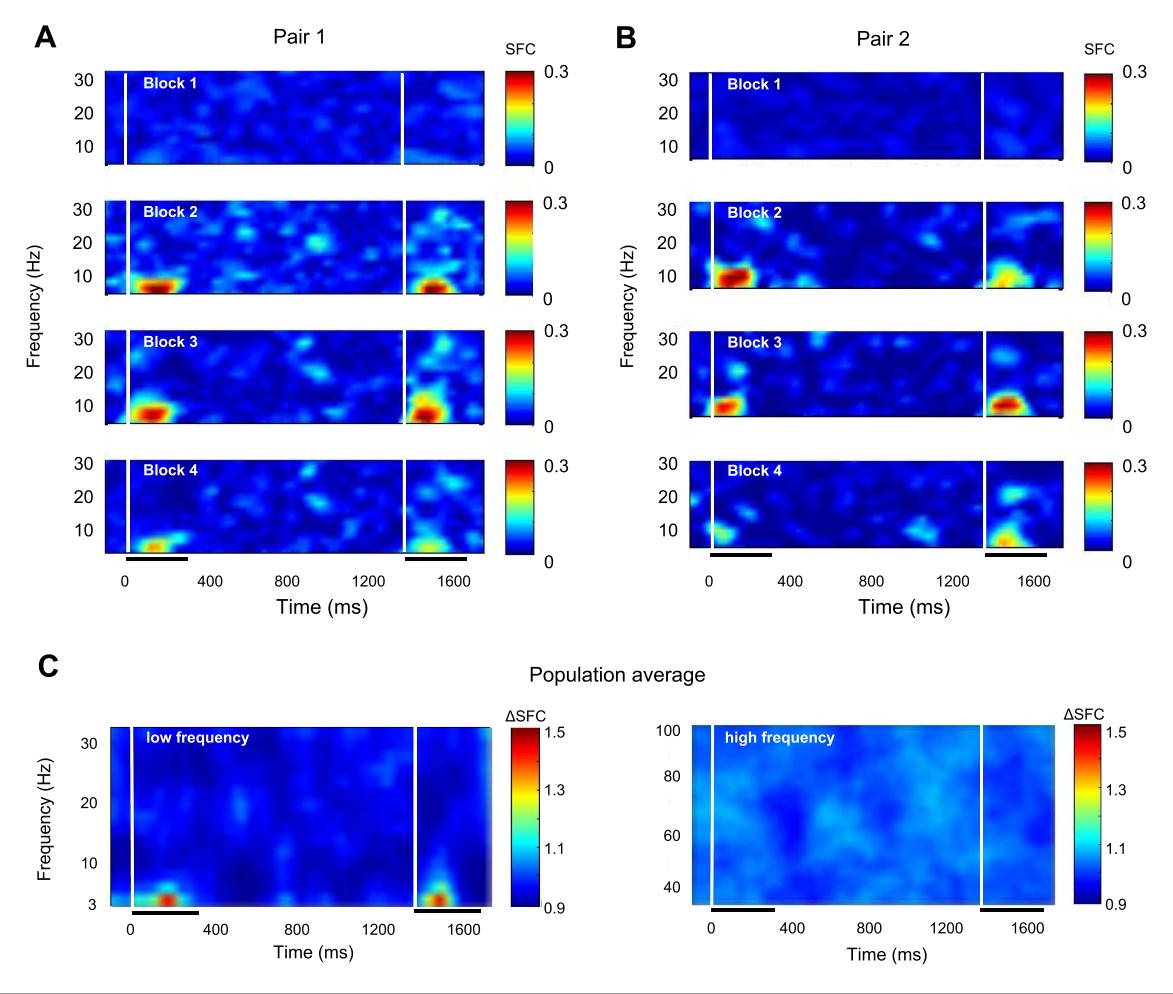

**Figure 2**. Rapid learning increases spike-LFP theta synchronization. (**A**, **B**) Spike-field coherence (SFC) from two example pairs of recording sites during blocks of learning. Each row shows the mean SFC in the low-frequency bands for the two example pairs in a particular block. (**C**) SPC—population average. The two panels show the population average (median change) of normalized SFC change in blocks 2–4 relative to block 1 throughout the trial. For each block, SFC was calculated within a 300-ms window sliding every 10 ms, and then the results were normalized for each session. The left panel shows SFC changes for the low frequencies, and the right panel represents frequencies within the gamma band. The x-axis represents time relative to the onset of the target stimulus. The two white vertical bars mark the onset of the target and test stimuli. The horizontal bars represent the time interval when the target and test stimuli are presented.

pairs), we found that the changes in theta SFC did not depend on the distance between electrode pairs (Wilcoxon signed-rank test, $p < 10^{-11}$ for both 'near' and 'far' pairs, see *Figure 3E,F*).

Our analysis in *Figure 2* suggests a sharp increase in theta spike-LFP coherence from block 1 to block 2 to match the changes in behavioral discrimination performance during learning (*Figure 1B*). However, since this analysis was performed on individual blocks of trials, this might have occluded a gradual trial-by-trial transition in theta SFC. To address this issue, we computed the changes in theta SFC and the behavioral discrimination threshold relative to the first 64 trials in the session using a sliding window of 64 trials (in 10-trial increments). As shown in *Figure 4*, there was a gradual increase in theta spike-LFP coherence across trials that matched the time course of the behavioral improvement during learning. This indicates that both the changes in theta spike-LFP coherence and the improvement in discrimination occur gradually during learning.

Previous work in area V4 has shown that working memory influences theta power and the phase synchronization between spikes and LFPs (*Lee et al., 2005*) and theta coupling between areas V4 and prefrontal cortex. To test the possibility that learning might be accompanied by a change in spike-LFP

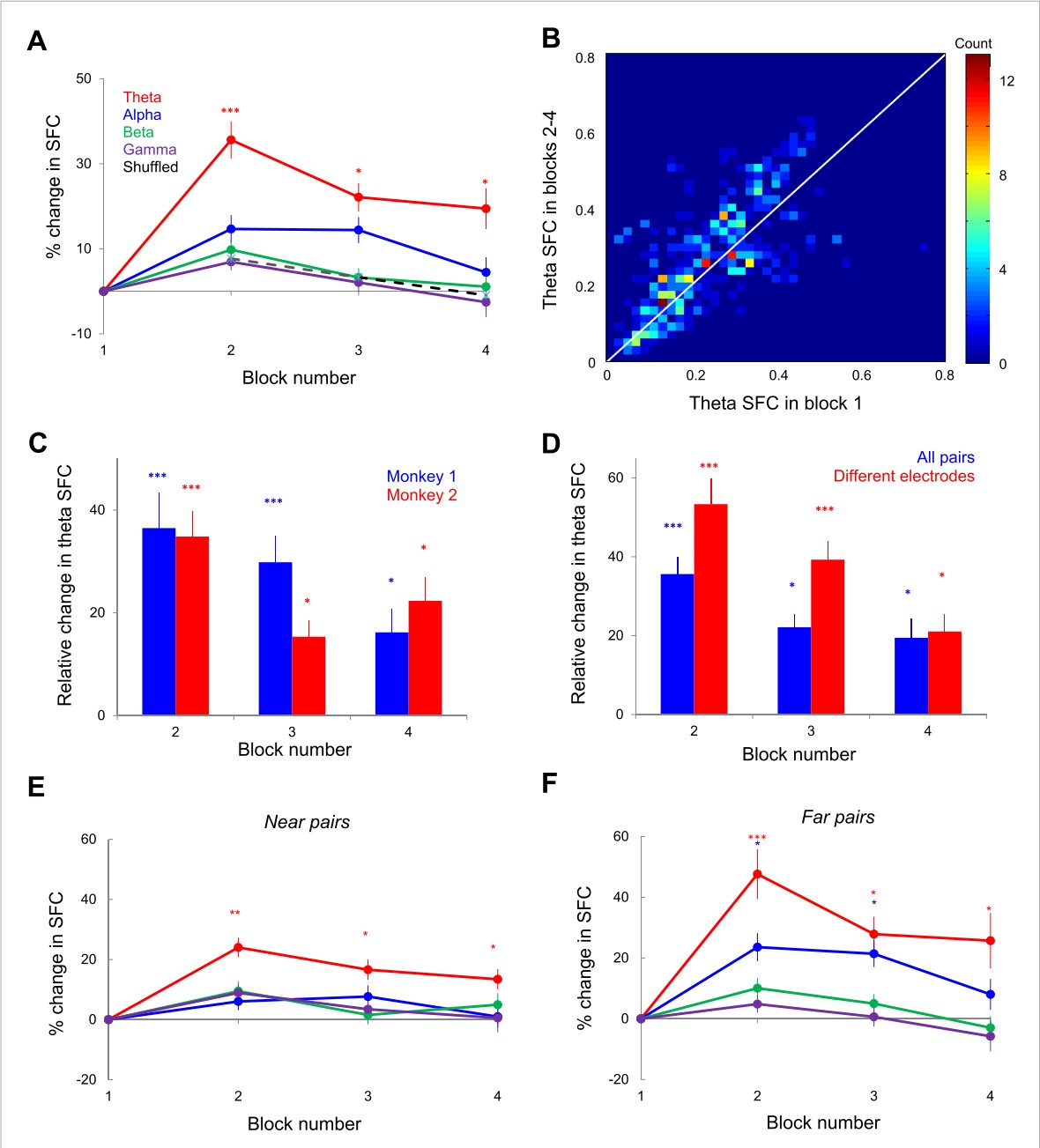

**Figure 3**. Changes in spike-LFP coherence during learning. (**A**) Relative change of mean SFC in each block and frequency band. The error bars represent s. e.m. across spike-LFP pairs. Asterisks for each point show the statistical significance of the difference between SFC in each of blocks 2, 3, and 4 relative to block 1 in each frequency band. (**B**) Distribution of mean theta SFC values in blocks 2–4 vs block 1. Color indicates the number of spike-LFP pairs measured in blocks 2–4 vs block 1 in 0.02 bins. (**C**) Block-by-block change in theta SFC relative to block 1 for each monkey. Error bars represent s.e.m. (* represents $p < 0.05$; *** represents $p < 0.001$). (**D**) Relative change in theta SFC when the pairs originating from the same electrode are included (blue, all pairs) or excluded (red, different electrodes). The error bars represent s.e.m. (**E**) Near pairs (electrode distance <2 mm): Kruskal–Wallis test showed no significant changes induced by learning in alpha, beta, and gamma bands ($p = 0.09$, 0.19, and 0.12, respectively), but a significant increase in theta band ($p = 2.29 \times 10^{-5}$). (**F**) Far pairs (electrode distance >2 mm): Kruskal–Wallis test showed no significant changes in alpha, beta, and gamma bands ($p = 0.58$, 0.10, and 0.13 respectively), but a significant increase in theta band ($p = 1.73 \times 10^{-6}$).

coherence, particularly in the theta band, we calculated SFC in the delay period between the target and test by dividing the delay into three time windows (early, middle, late), each of identical length to the stimulus period. However, we found that the median SFC in block 1 was not significantly different

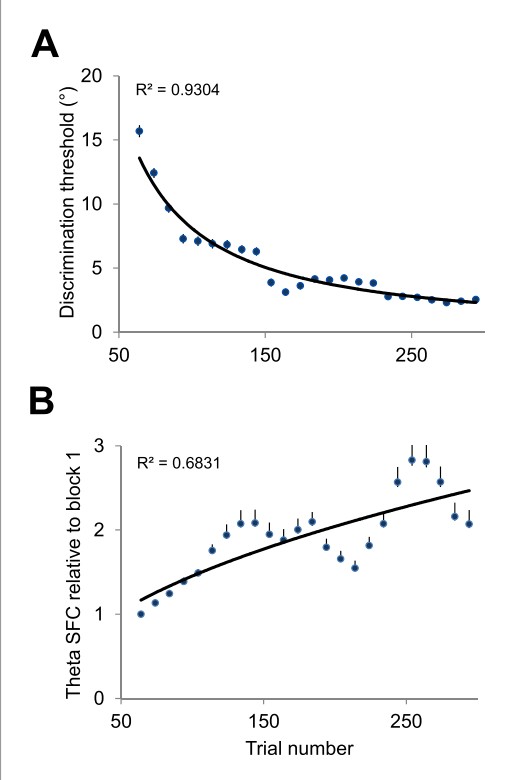

**Figure 4**. Gradual changes in theta spike-LFP coherence and behavioral performance during learning. (**A**) Mean behavioral discrimination threshold calculated throughout the session using a sliding window of 64 trials in steps of 10 trials. The solid line represents the exponential fit. The error bars represent s.e.m. (**B**) Median change in theta spike-LFP coherence (blocks 2–4 vs block 1) calculated throughout the session using a sliding window of 64 trials in steps of 10 trials. The solid line represents the exponential fit. The error bars represent the distance between the first and third quartiles divided by the square root of n (number of samples).

from that in blocks 2–4 in any of the windows of the delay period (*Figure 5*, p > 0.05, Wilcoxon signed-rank test for each delay interval). This indicates that the improvement in behavioral performance during learning is unlikely to be explained by a change in the working memory load when novel images are presented during training.

It is, in principle, possible that the low-firing neurons in our population may lead to measured SFC values biased toward high synchronization levels. To rule out this concern, for each neuronal pair, we examined the relationship between small changes in firing rate and theta SFC. However, we could not find a significant correlation between these two measures (r = 0.0178, p = 0.5711, Pearson correlation). We subsequently computed the z-scores of SFC values in each block (*Jarvis and Mitra, 2001*) by computing the distribution of coherence under null hypothesis given the number of degrees of freedom (i.e. the number of spikes * number of tapers), then calculating how many standard deviations the observed coherence differs from zero. The z-score eliminated the bias due to low-firing rates and the small sample size. We found that even when we compared the z-scored coherence values, theta SFC was still significantly elevated after block 1 (p < 0.05, Wilcoxon signed-rank test), whereas the z-score coherence in the other frequency bands either showed no change (in alpha and beta bands, p > 0.1) or decreased only slightly in the gamma band (p < 0.01). Altogether, these results further confirm that changes in theta SFC induced by learning are unrelated to changes in neuronal firing rates.

The analysis in *Figure 3* indicates that the increase in theta SFC is more prominent in block 2 when learning rate is the highest, followed by a decrease in blocks 3 and 4 when behavioral performance stabilizes (post hoc test, p < 0.05 for the decrease in theta SFC in block 3 vs 2; p < 0.001 for block 4 vs 3). This raises the possibility that the increase in spike-LFP theta coherence may only be required during the development of learning, but not after the behavioral threshold has reached the asymptote (following block 2). To test this possibility, we performed a new set of experiments in which the same image was presented in two consecutive sessions. Since after the first session the image became familiar (behavioral performance was improved at the end of the session), this offered us the opportunity to examine the changes in theta spike-LFP coherence for novel and familiar images. Indeed, for familiar image sessions (n = 23), we found that, as expected, the orientation discrimination threshold was <5° in block 1 and performance did not improve further during the session (p > 0.5 for each block comparison; ANOVA test, *Figure 6A* inset). Importantly, in contrast to novel images, the sessions in which we presented familiar images were not associated with significant changes in spike-LFP coherence in any frequency band (*Figure 6A*; p > 0.05, Kruskal–Wallis test, comparing SFC in block 1 vs blocks 2–4; n = 625 pairs).

One important control is to ensure that the changes in theta SFC observed during rapid learning across blocks of trials are not due to the stimulus presentation itself. We thus collected data from one animal in which five control sessions were recorded before the monkey was able to accurately perform

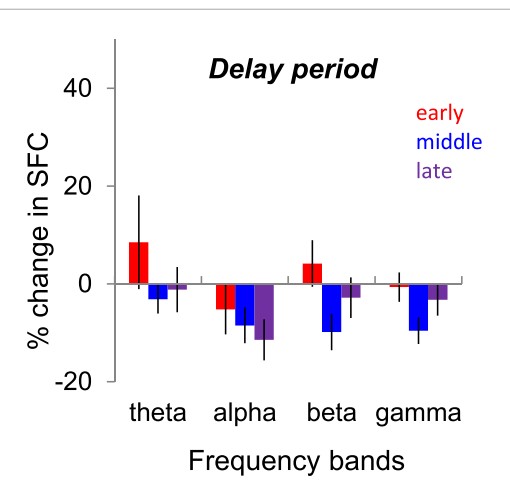

**Figure 5**. Changes in spike-LFP coherence (SFC, spike-field coherence) during the delay period. Median change in SFC (blocks 2–4 vs block 1) during the delay period for each frequency band. The 1000-ms delay period was divided into three 333-ms windows referred as early, middle, and late. *** denotes p < 0.001; ** denotes p < 0.01.

the behavioral task with the set of 10 prototype stimuli. However, despite the fact that animals performed a discrimination task identical to that described in *Figure 1A* and novel stimuli were presented in the same conditions, learning did not take place. Indeed, as shown in *Figure 6A* (top), for the 'no learning' control sessions, behavioral discrimination threshold was high at the beginning of the session (in block 1) and did not undergo statistically significant changes in subsequent blocks (p > 0.1, ANOVA test). Importantly, examining the changes in spike-LFP coherence for the population of 96 pairs, we found that SFC did not change across blocks of trials in any frequency band (p > 0.1, Kruskal–Wallis test, comparing SFC in block 1 vs blocks 2–4). Similar results (no statistically significant block-by-block changes in SFC in any frequency band, *Figure 6—figure supplement 1*) were found when natural images were flashed in the neurons' receptive fields during passive (inattentive) fixation experiments (n = 12 sessions, 741 pairs). Altogether, these results indicate that the coordination of spike timing with the local theta LFP activity only occurs when animals improve their performance during learning, and that the increase in theta SFC does not continue with subsequent exposure to familiar stimuli after learning has stabilized, or in conditions in which learning does not take place.

Finally, we examined whether the session-by-session increase in spike-LFP synchronization is related to the improvement in behavioral performance during learning. Thus, we measured the correlation between the average change in theta SFC across all the pairs recorded in a given session (blocks 2–4 vs block 1) and the corresponding change in behavioral discrimination threshold (*Figure 6B*). We found that the decrease in behavioral discrimination threshold is significantly correlated with the increase in theta SFC (r = −0.49, Pearson correlation, p < 0.01). In contrast, despite slightly negative trends, there was no significant correlation between the change in SFC and behavioral performance in the higher frequency bands (alpha: p = 0.42, beta: 0.94, and gamma: 0.17; *Figure 6C*).

## Discussion

We have demonstrated that the increase in behavioral discrimination performance during learning is associated with a tight coordination of spike timing with the local population activity. The spike-LFP synchronization associated with learning occurred in the theta band, while higher frequency synchronization was uncorrelated with changes in behavioral performance. Importantly, the changes in spike-LFP theta synchronization only characterized the learning of novel stimuli but were absent once stimuli became familiar or in situations when learning did not occur. These findings indicate that plasticity in visual cortex during the time course of learning is accompanied by elevated low-frequency synchronization between individual neuron responses and local population activity. Additionally, there was a significant increase in spike-LFP phase locking strength in the theta band during learning despite the fact that theta LFP power remained constant. Although our study reports neuronal changes associated with a rapid form of learning, it is possible that the improvement in behavioral performance occurring over larger time scales (e.g., weeks) may be accompanied by similar changes in spike-LFP theta synchronization.

Our results are consistent with the fact that learning induces rapid changes in the strength of synapses. Indeed, synaptic plasticity has been associated with coordinated action-potential timing across neuronal networks and oscillations of specific frequencies. In particular, neuronal oscillations in the theta frequency range (3–8 Hz) have been associated with the induction of synaptic plasticity (*Buzsaki, 2002*). For instance, the timing between incoming stimuli and ongoing theta oscillations

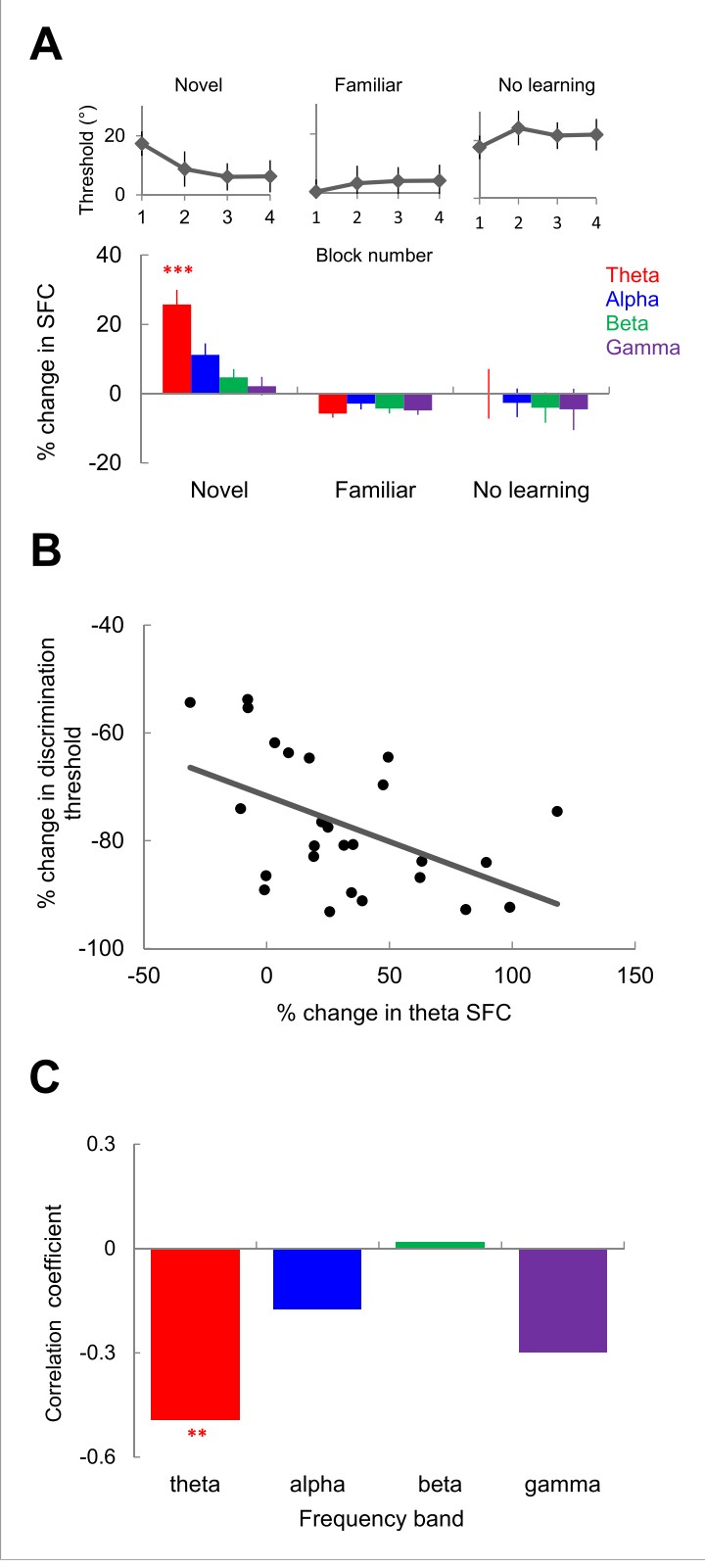

**Figure 6**. Relationship between spike-field theta synchronization and behavioral performance. (**A**) Relative change of SFC in different frequency bands during exposure to novel images (left), during exposure to familiar images (middle), and in the 'no learning' condition (right). Error bars represents s.e.m. across spike-LFP pairs. The inset on top shows block-by-block behavioral performance in each type of session. (**B**) Correlation between the session-by-session change
*Figure 6. continued on next page*

*Figure 6. Continued*

in monkey's orientation discrimination threshold and the mean change in theta synchronization in blocks 2–4 vs block 1 (by averaging across the spike-LFP pairs in a given session). The dark line represents the linear regression fit. (**C**) Correlation coefficient between the session-by-session change in behavioral discrimination threshold and SFC in different frequency bands. Double asterisks show statistical significance (p < 0.01) in theta band.

The following figure supplement is available for figure 6:

**Figure supplement 1**. Control experiment—(**A**) Monkey 2 performed control experiments (n = 12 sessions, 741 cell pairs) in which novel natural scenes were flashed in the neurons' receptive fields, while the animal was engaged in a red/green color detection task in the contralateral hemifield.

controls synaptic changes. In addition, oscillations in the theta range can be modulated by behavior and brain state (*Winson, 1978*; *Hasselmo et al., 2002*) and have been assigned a key role in the maintenance of information in short-term memory (*Lee et al., 2005*; *Liebe et al., 2012*). Although our study did not reveal an ongoing theta oscillation for the population activity, we demonstrate that the timing of neuronal spikes relative to the theta-filtered LFP activity during a rapid form of visual learning is correlated with the increase in behavioral performance. Our results are consistent with previous evidence in rabbit hippocampus (*Berry and Thompson, 1978*) that animals that exhibit more theta activity show elevated classical conditioning of the nictitating membrane response.

The increase in spike-LFP theta synchronization during learning is unlikely to be explained by changes in behavioral context, such as gradual changes in attention or alertness as the session progresses. Indeed, although attention and alertness have been previously associated with elevated neuronal firing in area V4 (*Fries et al., 1997*; *McAdams and Maunsell, 1999*; *Fries et al., 2001*), increased neuronal sensitivity (*McAdams and Maunsell, 1999*), increased gamma LFP power (*Fries et al., 1997*, *2001*), and increased gamma spike-LFP synchrony (*Fries et al., 1997*; *Jarvis and Mitra, 2001*), none of these measures changed significantly across blocks of trials (p > 0.1, for both animals). We also failed to find changes in eye movements during learning—there was no significant relationship between the horizontal/vertical saccade amplitude and frequency and block number (p > 0.4, Pearson correlation for each comparison). This indicates that the increase in theta coherence during learning is unlikely to have been contaminated by gradual changes in behavioral context or fixational eye movements during the time course of the session.

In principle, it may be possible that the behavioral improvement in blocks 2, 3, and 4 could reflect a change in monkey's strategy to respond, for instance, by frequently holding the lever after block 1 until the end of the session, rather than learning to perform the image discrimination task. To rule out this possibility, we examined the block-by-block changes in behavioral performance in the match trials (these trials require a bar release response). If monkey's strategy was to keep holding the lever as the session progressed, performance in the match trials (randomly interleaved with the non-match trials) would significantly deteriorate. However, we did not find statistically significant changes in match (bar release) responses across blocks of trials (by analyzing all the sessions with an improvement in learning performance, p > 0.1; ANOVA test). Another possibility that could explain our behavioral results is that the poorer performance at the beginning of the session (in block 1) may be due to animals being distracted by the novelty of stimulus presentation. To rule out this concern, we performed additional behavioral experiments (Monkey 2, n = 10 sessions) in which the animal was passively exposed to a novel natural scene while he performed a color detection task (red–green task) in the contralateral hemifield (*Figure 1—figure supplement 1*). After 150 trials of passive (inattentive) exposure to the novel image, the monkey was switched to the image discrimination task (*Figure 1A*). As expected, as the monkey did not actually practice the image orientation discrimination task for the stimuli he was exposed to, we found a gradual improvement in behavioral discrimination threshold during the session (p < 0.01, ANOVA test; each of blocks 2–4 was characterized by a lower threshold than block 1, p < 0.05, post hoc multiple comparisons). This control was repeated with images that were fixated (without controlling attention), but the results were similar.

Several possible mechanisms could modulate spike-timing accuracy in V4. Indeed, it is possible that theta oscillations in areas outside V4 may modulate excitability in areas projecting to and receiving information from V4. For instance, theta synchronization was associated with pattern recognition,

working memory, sequence learning and navigation, most prominent in the temporal lobe (*von Stein et al., 2000*; *Lengyel et al., 2005*; *Rutishauser et al., 2010*; *Hoerzer et al., 2010*; *Benchenane et al., 2010*). It also has been suggested that low-frequency synchronization is suitable for long-range or polysynaptic communication across distant brain areas (*von Stein et al., 2000*). One possibility is that visual cortical neurons are coordinated with the theta rhythm in higher cortical areas, for example, activation of dopaminergic neurons of the ventral tegmental area or those in prefrontal cortex influences theta phase-locking of neurons (*Benchenane et al., 2010*). Both of these types of neurons can be activated by novel stimuli, similar to those used in our experiments. In addition, theta synchronization within and between V4 and prefrontal cortex has been reported during the maintenance of visual short-term memory (*Raghavachari et al., 2001*; *Liebe et al., 2012*). The fact that we found a significant correlation between the accuracy of theta phase-locking during learning and behavioral performance suggests a relationship between synaptic plasticity in downstream areas and efficient information flow in visual cortex to facilitate learning.

## Materials and methods

All experiments were performed in accordance with protocols approved by the U.S. National Institutes of Health Guidelines for the Care and Use of Animals for Materials and methods and were approved by the Institutional Animal Care and Use Committee at the University of Texas Health Science Center at Houston.

### Surgical procedures for recording chamber and head-holder implantation in monkey

All surgical procedures were overseen by UTHSC-H Animal Welfare Committee. A titanium head post was implanted in medial frontal region with the help of multiple anchor screws. Following a recovery period of about 10 days, monkeys were trained for 3–4 months on visual fixation and discrimination tasks. After the monkey learned the tasks, a recording chamber (inner diameter of 19 mm) for single unit multiple electrode recording was cemented over area V4 (according to MRI map). A few stainless steel screws were inserted into the skull around the recording chamber and a thin stainless steel wire was wrapped around the screws for additional support.

### Behavioral paradigm

Two male rhesus monkeys (Macaca mulatta) were trained to perform a natural image orientation discrimination task. Stimuli were presented on a CRT color monitor (Dell, Texas, United States, 60-Hz refresh rate, running MATLAB and using Psychophysics Toolbox), positioned 57 cm in the front of the animal. After monkeys triggered the trial by holding a bar, a small spot (0.1 deg) was presented in the center of monitor. Monkeys were required to hold fixation within a 1-deg diameter window throughout stimulus presentation; the trial was automatically aborted if fixation instability (microsaccade amplitude) exceeded 0.25 deg at any time during stimulus presentation. All natural scenes were converted to equal-contrast gray scale circular images of 5 deg in diameter. After 500 ms of fixation, a target image was flashed for 300 ms. After a 1000-ms blank, a 300-ms test stimulus of random orientation (rotated with respect to the target image by 0, 3, 5, 10, or 20°) was flashed at the same visual location. The monkey was required to hold the bar for 1.5 s if the target and test stimuli were different and release the bar within 1.5 s if they are identical. The monkey was rewarded with 5 drops of juice for the correct choice. A full session consisted of 4 consecutive blocks of 96 trials each. Test image orientation was randomized across trials in each block (each block was randomly composed by 48 match 12 non-match trials for each image orientation). Psychometric curves were obtained in each session. We averaged the percentage correct responses for each orientation difference between target and test. The probability of false alarms (which is the value obtained for the 0° orientation difference) represents the proportion of match trials in which the monkey's response was incorrect. The psychometric curves represent the best fit of the data using Weibull functions: $P(\Delta\theta)=1-(1-FA) \exp-(\Delta\theta/a)^b$, where $FA$ is the false alarm rate, and $a$ and $b$ are the offset and slope terms of the best Weibull fit. The threshold was computed as the orientation difference at which accuracy is 75% (the threshold takes into account the false alarm rate).

To rule out that the behavioral improvement in blocks 2, 3, and 4 could be due to a change in monkey's strategy to respond (i.e. by repeatedly holding the bar), we examined the block-by-block changes in performance in the match trials (these trials required a bar release response). If monkey's

strategy was to keep holding the lever as the session progressed, performance in the match trials (randomly interleaved with the non-match trials) would significantly deteriorate. However, we did not find statistically significant changes in match (bar release) responses across blocks, by analyzing all the sessions in which found an improvement in learning performance (p > 0.05; ANOVA test).

Stimulus presentation and eye position monitoring was manipulated and synchronized with neuronal data using the ECM (Experiment Control Module) programmable device (FHC Inc). Eye position was continuously monitored using an eye tracker system (EyeLink II, SR Research Ltd., Osgoode, ON, Canada) that offers a binocular 1-kHz sampling rate. Eye position was calibrated before each experiment using a 5-point calibration procedure in which the animal was required to fixate on each one of 5 points (1 in the center, 2 in the vertical, and 2 in the horizontal axes or the diagonals) in steps of 4, 8, and 12 deg from the central fixation spot. We analyzed the eye position on the x- and y-axis, as well as the number and speed of microsaccades. The eye-tracker gains were adjusted such as to be linear for the horizontal and vertical eye deflections. The fixation pattern was carefully analyzed offline. Microsaccades were analyzed every 10 ms by using a vector velocity threshold of 10 deg/s (this corresponds to a 0.1 deg eye movement between consecutive 10-ms intervals). If a detected microsaccade exceeds 0.25 deg (fixation instability), the trial is automatically aborted.

## Data acquisition

We used two types of electrode systems in each monkey: (i) arrays of parylene-C-coated tungsten microelectrodes (MPI, 1–2 MΩ at 1 KHz) grouped in pairs and attached to several micro-drives (Crist) fixed on a grid, and penetrated transdurally through stainless steel guide tubes into the cortex; (ii) 16-channel U-probes (Plexon) with contacts spacing at 100 μm advanced using the NAN drive system (Plexon) attached to the recording chamber. In each session, we advanced up to 8 tungsten microelectrodes and/or 2 U-probes into area V4. Real-time neuronal signals from multiple channels (up to 32, simultaneous 40 kHz A/D conversion on each channel) were recorded and processed through Multichannel Acquisition Processor system (MAP, Plexon Inc). The signals were first filtered by a preamplifier box into spike channels (150 Hz–8 kHz, 1 pole low-cut, 3 pole high-cut, with programmable referencing, 50× gain) and field potential channels (0.07, 0.7, 3–170, 300, 500 Hz user selectable, 1 pole low-cut, 1 pole high-cut, 50×). Single-unit signals were further amplified, filtered, and viewed on an oscilloscope and heard through a speaker. The spike waveforms above threshold were saved and fine sorted after data acquisition was terminated using Plexon's offline sorter program. After a unit was isolated, its receptive field was mapped with dynamic gratings or using reverse correlation while the animal maintained fixation. As a measure of neuronal discrimination performance, we calculated the neurons' capacity to discriminate between each combination pair of test orientations (d', *Green and Swets, 1966*), calculated by dividing the absolute difference between the neurons' mean firing rates by the rms standard deviation).

## LFPs

Low-frequency field potential signals were amplified and digitized at 1 kHz. To correct for the time delays induced in the LFP signals by the filters in headstage and pre-amplification board, we used the software correction FPAlign provided by Plexon (http://www.plexon.com/downloads.html). LFPs were further filtered between 0.5 Hz and 100 Hz using a fourth order Butterworth filter. In order to remove line artifacts, we applied a digital notch at 60 Hz (fourth order elliptic filter, 0.1 db peak-to-peak ripples, 40 db stopband attenuation). All filtering was applied by using forward and backwards filtering to obtain zero phase shifts. We discarded all LFPs that had more than 3 points outside the mean ±4 standard deviations to avoid influence of irregular artifact noise from muscle activity or other sources. The amplitude of LFPs was measured in each trial by standard deviation, peak-valley amplitude, and average voltage for specific periods of interest. We estimated the LFP power density during the presentation of the stimuli using sliding windows of ±150-ms length in steps of 10 ms. In order to obtain optimal spectral concentration, we used the multitaper method by multiplying each data epoch with the tapers before Fourier transforming (*Mitra and Pesaran, 1999*; *Jarvis and Mitra, 2001*; *Pesaran et al., 2002*; *Womelsdorf et al., 2006*). The number of tapers was calculated according to the formula: $K = 2 TW - 1$, where $K$ is the highest number of tapers that can be used while preserving optimal time-frequency concentration of the data windowing available from the Slepian taper sequences, $T$ is the length of the data in seconds, and $W$ is the half-bandwidth of the multitaper filter. For our analysis, we applied spectral smoothing of $TW = 4$, $K = 7$

tapers for frequencies greater than 30 Hz and a single Hanning taper for lower frequencies. The power spectral density was first normalized to the average power spectrum during the 300-ms fixation period before the presentation of the target stimulus (averaged across all trials in a session). This helped balance the power spectrum between low and high frequencies within same amplitude range. The average power of theta, alpha, beta, and gamma frequency bands was calculated as the mean power at frequencies between 5 and ~7 Hz, 8–13 Hz, 15–30 Hz, 35–80 Hz. To compare LFP power at individual channels across session, we further normalized the power in each block and frequency band to its mean value in the first block.

## SFC

To examine whether learning increases the coupling between the spike trains and LFPs, we calculated the STA and SFC. The coherence between two signals is a complex quantity whose magnitude is a measure of the phase synchrony for a given frequency. We computed STA by averaging the LFP signal within a window centered ±150 ms on each elicited spike. The 300-ms window enables us to measure the spectrum of the STA over entire theta frequency range. To quantify STA, we calculate its power spectrum (i.e. the magnitude of all frequency components of the STA as a function of frequency). SFC was computed by dividing the power spectrum of the STA by the average of all power spectra of the LFP segments that were used to obtain the STA (*Fries et al., 1997*; *Womelsdorf et al., 2006*). Thus, SFC is independent of the firing rate of the single units and the power spectrum of the LFPs. SFC ranges from 0, lack of synchronization, to 1, perfect phase synchronization. We assessed the temporal SFC spectrum by using windows of ±150 ms that were moved over the data in 10-ms steps from 100 ms before the onset of the target stimulus to 100 ms after the disappearance of the test stimulus.

To eliminate the onset transient response that could cause artifacts in the computation of SFC, we focused our analysis in the 150–350-ms window from the onset of each stimulus. To eliminate the bias caused by finite data set and eliminate the block-by-block bias in the calculation of SFC (caused by the different number of spikes in each block), we randomly sampled the stimulus interval to extract the same number of spikes in each block, and then used these spikes to calculate the STA (for each pair of electrodes). We also computed the z transformed SFC. It was defined by formula $z = \beta^*(\sqrt{-(V-2)^*\log(1-|C|^2)} - \beta)$ , where $\beta = 1.15$, $C$ is the SFC, and V is number of degrees of freedom. Under null hypothesis, the z-transformed SFC value is distributed as a normal variate with variance equal to 1. The z-scored SFC indicates how many standard deviations the observed SFC differs from zero.

To validate our calculation of SFC, we used the Chronux function *coherencycpt*, which computes the multitaper SFC for a continuous signal (LFP) and point process data (spike-train) according to an optimal family of orthogonal tapers derived from Slepian functions. However, our results were identical when SFC was calculated using the STA method or the Chronux function *coherencycpt*.

To calculate the alignment between spikes and LFPs, we computed the LFP phase in theta band for each spike. The LFP signal was filtered in the theta band by an equiripple FIR filter (band edge 3.5–4 Hz, 8–8.5 Hz, attenuation 40 dB, 1 ripple). Similar filtering was applied for other frequency bands (alpha 7–8 Hz, 14–15 Hz; beta 13–14 Hz, 30–31.5 Hz; gamma 28–30 Hz, 80–82 Hz). The filtered data were subsequently Hilbert transformed. The phase at each time was defined as the angle of the Hilbert complex. We analyzed the distribution of phase angles at each spike time within 150–350 ms after stimulus onset. Circular statistical analysis was performed using the Matlab CircStat toolbox. The significance of phase angle distribution non-uniformity was assessed by Rayleigh's and Omnibus tests at both individual pair and population level. Rayleigh's Z score is a measurement of the strength of non-uniform circular distribution. It was defined as the square of vector sum of all sample angles divided by the sample size. The difference in phase distribution was assessed for statistical significance by using the non-parametric circ_cmtest through all blocks and then Kuiper test between block pairs. To verify that the phase locking between spikes and LFPs in the theta band was not due to the stimulus itself, we calculated the theta phase of shuffled control. It was performed by shuffling the trials of LFP data to match original spike trains. Each trial was sampled once. We calculated the Rayleigh's Z value of shuffled spike-LFP phase. The median of 100 different reshuffle was considered as the shuffled Z value.

## Acknowledgements

We thank Sarah Eagleman and Sorin Pojoga for help with animal training and experimental programming. Supported by the NIH EUREKA Program, the National Eye Institute, and Texas ARP Program (VD).

## Additional information

### Funding

| Funder | Author |
| --- | --- |
| NIH EUREKA Program | Valentin Dragoi |
| National Eye Institute (NEI) | Valentin Dragoi |

The funders had no role in study design, data collection and interpretation, or the decision to submit the work for publication.

### Author contributions

YW, Acquisition of data, Analysis and interpretation of data, Drafting or revising the article; VD, Conception and design, Drafting or revising the article

### Ethics

Animal experimentation: This study was performed in strict accordance with the recommendations in the Guide for the Care and Use of Laboratory Animals of the National Institutes of Health. All of the animals were handled according to approved institutional animal care and use committee (IACUC) protocols (AWC-14-0114) of the Texas Health Science Center at Houston. The protocol was approved by the Committee on the Ethics of Animal Experiments of the Texas Health Science Center at Houston. All surgery was performed under isoflurane anesthesia, and every effort was made to minimize suffering.

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
