## [Decision Letter]

Thank you for submitting your work entitled “Rapid learning in visual cortical networks” for peer review at *eLife*. Your submission has been favorably evaluated by Timothy Behrens (Senior Editor), and three reviewers, one of whom, Sacha Nelson, is a member of our Board of Reviewing Editors. One of the three reviewers, Alfredo Kirkwood, has also agreed to share his identity.

The reviewers have discussed the reviews with one another and the Reviewing Editor has drafted this decision to help you prepare a revised submission.

The authors analyze coherence between firing of individual neurons and a field potential measure of population activity and find that the coherence increases as an animal learns to perform a perceptual task. The significance is high as it shifts the emphasis from the single unit level to the population level. Furthermore, the authors show that the coherence only occurs within a particular frequency range (theta) previously associated with learning in multiple brain structures.

Essential revisions:

All three reviewers agreed that the case for the core conclusions would be substantially strengthened by experiments that showed that the spike-field coherence does not occur under conditions in which learning does not occur – i.e. they are not a product of the stimulus itself. The reviewers agreed that while the existing demonstration of a plateau in learning and coherence was helpful it was not sufficient. One reviewer suggested that analysis of error trials might be helpful in sorting this out.

There was also general agreement that the authors should take care to indicate that they have established correlation but not causality and that both directions of causality (coherence causing learning or learning causing coherence) have not been ruled in or out.

Reviewer #1:

The present study makes the interesting observation that perceptual improvement in an object orientation matching task is associated with improved spike-field coherence in monkey V4 at theta, but not other frequency ranges, and in the absence of large changes in firing or tuning of single units, or in the power of the underlying field potentials.

1) The major limitation to the present study is that the authors have not shown that similar stimuli do not produce increased coherence in the absence of learning. They have performed experiments that show that once learning has occurred and the same stimuli are presented again no further increases in coherence are observed and no further increases in performance occur. In addition the authors mention an additional behavioral control in the Discussion but it does not appear that this data is shown or otherwise referred to in the text. The ideal control would be to show the same stimuli under conditions in which the animal is not improving performance (e.g. because they are not doing a task or are doing a different task) and test whether or not the observed network changes in coherence occurred or not.

2) A second concern is that the sensitivity to coherence may differ for different frequency ranges irrespective of learning induced changes e.g. in Figure 1 theta power is constant but other bands decline in power with block – this effect looks significant. There could also be effects related to the number of action potentials falling within a period at each frequency. I suspect that the normalization procedures used circumvent this, but it is not easy for the reader to be sure of this. The authors should make a better case that their analysis is equally sensitive to coherence at different frequency ranges.

Reviewer #2:

The article by Wang and Dragoi shows that perceptual learning during a task that can be learned in a single experimental session is correlated with spike-field coherence between single neurons and the theta band. The authors speculate that the spike-field coherence might be part of the mechanism responsible for the increase in behavioral performance with learning. The phenomenon that they discover is novel, as is the idea of recording from cells as learning occurs during a single session.

1) The data and analysis seem solid, but I have significant concerns about the “marketing” of the results. The authors imply in many places that it is somehow more likely that the spike-field coherence is responsible for the learning than a result of the learning. I don't think the authors can or should say more than state the fact that these two features co-occur and are correlated with one another. I don't think it makes the paper more or less interesting if the speculative idea that spike-timing-dependent plasticity is underlying the learning, and that one might expect to see more synchronized activity when spike-timing-dependent plasticity is occurring.

2) The only specific requirements I have are to re-write some of the paper so that it is clear that the authors are only establishing a correlation between these two quantities. The paper reads as though there is some evidence for causation in a particular direction (that the SPC is causing the learning).

Examples:

Abstract: the word “predicted”.

Results, fourth paragraph: “clearly, rapid learning causes” (should be co-occurs).

Results, fifth paragraph: “learning-induced theta SFC” (occurs twice), implies the learning induced the theta SFC, “learning-induced” occurs at least once.

Results, tenth paragraph: “coordination of spike timing with the local theta LFP activity is only required during the improvement in behavioral performance during learning” should say “only co-occurs during…” or something similar.

Reviewer #3:

This is a very interesting study linking performance in a perceptual learning task with spike “synchronization” with the LFP in the theta range. The significance is high as it shifts the emphasis from the single unit level to the population level. The data and analysis seems convincing (although I am an expert in this type of quantitative approaches). I have only one general consideration to raise, that the rationale for doing the studies in V4 is not clear. This is important because many forms of perceptual learning do take place in V1, raising the possibility that observation in V4 are secondary to other changes occurring at earlier stages of perception. Perhaps the reason for choosing V4 is obvious, but it needs to be stated.

[Editors' note: further revisions were requested prior to acceptance, as described below.]

Thank you for submitting your work entitled “Rapid learning in visual cortical networks” for peer review at *eLife*. Your submission has been favorably evaluated by Timothy Behrens (Senior Editor) and three reviewers, one of whom, Sacha Nelson, is a member of our Board of Reviewing Editors. One of the three reviewers, Alfredo Kirkwood, has also agreed to share his identity.

The reviewers have discussed the reviews with one another and the Reviewing Editor has drafted this decision to help you prepare a revised submission.

The two reviewers are satisfied but the Reviewing Editor wishes you to clarify the following point. This should be very quick to address:

1) Why did learning fail to occur during five additional sessions? What was different about these sessions than the others illustrated? (This should be made clear in the manuscript).

---

## [Author Response]

*Essential Revisions*:

*All three reviewers agreed that the case for the core conclusions would be substantially strengthened by experiments that showed that the spike-field coherence does not occur under conditions in which learning does not occur – i.e. they are not a product of the stimulus itself. The reviewers agreed that while the existing demonstration of a plateau in learning and coherence was helpful it was not sufficient. One reviewer suggested that analysis of error trials might be helpful in sorting this out*.

We performed two additional control experiments to rule out that theta spike-field coherence increases as an effect of repeated stimulus presentation during the time course of the experiment.

First, we collected data from five additional sessions in which novel stimuli were presented, but learning did not occur, despite the fact that the animal performed an image discrimination task identical to that described in Figure 1 (there was no improvement in discrimination performance across blocks of training). Indeed, as shown in Figure 6 (top), for the ‘no learning’ control sessions, behavioral discrimination threshold was high at the beginning of the session (in block 1), and did not undergo statistically significant changes within subsequent blocks (P > 0.1; ANOVA test). Importantly, examining the changes in spike-LFP coherence for the population of 96 pairs, we found that SFC did not change across blocks in any frequency band (P>0.1, Kruskal-Wallis test, comparing SFC in block 1 vs. blocks 2-4).

Second, we performed additional control experiments (n=12 sessions, 741 cell pairs) in which novel natural scenes were flashed in the neurons’ receptive fields while the animal was engaged in a red/green color detection task in the contralateral hemifield (Figure 6—figure supplement 1s). Briefly, after 500 ms of fixation, a 5-deg natural scene and a red square (3 deg in diameter) were presented simultaneously for the same random duration (1000-1800 ms) at two symmetric locations on the screen. The animal was required to signal the color change for the attended square (from red to green) within the next 3000 ms. Each session consisted of 400 trials.

This experiment allowed us to examine the block-by-block (each block consisted of 100 trials) changes in spike-LFP coherence (SFC) when the image was presented in the neurons’ receptive fields (during the first 1000-ms of stimulus presentation). The results (shown as SFC changes with respect to SFC in block 1) are shown in Figure 6—figure supplement 1 (right panel) and confirm the results of our first control experiment, i.e., we were unable to detect significant block-by-block changes in SFC in any frequency band during inattentive (passive) fixation (theta: P=0.99, alpha: P=0.21, beta: P=0.25, gamma: P=0.49, Kruskal-Wallis test).

*There was also general agreement that the authors should take care to indicate that they have established correlation but not causality and that both directions of causality (coherence causing learning or learning causing coherence) have not been ruled in or out*.

We have revised the text throughout the manuscript to clearly indicate that learning is *associated* (or *correlated*) with changes in spike-field coherence. Although the original manuscript never claimed that learning and coherence changes are causally related, we have eliminated expressions like “learning-induced” or “changes in coherence predict learning” from the revised manuscript.

Reviewer #1:

*1) The major limitation to the present study is that the authors have not shown that similar stimuli do not produce increased coherence in the absence of learning. They have performed experiments that show that once learning has occurred and the same stimuli are presented again no further increases in coherence are observed and no further increases in performance occur*.

This issue was addressed in our response to Essential revisions above.

*In addition the authors mention an additional behavioral control in the Discussion but it does not appear that this data is shown or otherwise referred to in the text. The ideal control would be to show the same stimuli under conditions in which the animal is not improving performance (e.g. because they are not doing a task or are doing a different task) and test whether or not the observed network changes in coherence occurred or not*.

The data for the behavioral control experiment mentioned in the Discussion section is shown in Figure 1—figure supplement 1. Monkey 2 was passively exposed for 10 sessions to novel natural scenes (similar to those in the learning experiments reported in the manuscript) while the animal had to perform a color detection task (red-green task) in the contralateral hemifield. After 150 trials of passive (unattended) exposure to the image, the monkey was engaged in the rapid learning experiment described in the manuscript (Figure 1). However, even though images themselves were familiar to the animal, the fact that the monkey did not actually practice the image orientation discrimination task led to behavioral effects similar to those reported in the manuscript. That is, we found a gradual improvement in discrimination threshold at the end of each session – the gradual learning curve looked similar to that obtained in the original experiments. These results are shown in Figure 1—figure supplement 1 (asterisks indicate P<0.05).

*2) A second concern is that the sensitivity to coherence may differ for different frequency ranges irrespective of learning induced changes e.g. in*
Figure 1
*theta power is constant but other bands decline in power with block – this effect looks significant. There could also be effects related to the number of action potentials falling within a period at each frequency. I suspect that the normalization procedures used circumvent this, but it is not easy for the reader to be sure of this. The authors should make a better case that their analysis is equally sensitive to coherence at different frequency ranges*.

We believe that our analysis is equally sensitive to coherence at different frequency ranges. The spike-triggered average (STA) used in the calculation of SFC is measured by summing all LFP segments and then dividing by the number of spikes. Even though the power spectrum of the STA depends on the power spectrum of the LFP signal (decreasing LFP amplitude decreases the STA power despite the absence of spike-LFP synchronization), SFC is obtained by normalizing the power spectrum of the STA by the average of all power spectra of all LFP segments that were averaged to obtain the STA. This normalization ensures that SFC is independent of the spiking rates and the power spectrum of the LFP. Thus, the small block-by-block changes in LFP power in the alpha, beta, and gamma bands (Figure 1) are unlikely to alter the sensitivity of our SFC measure during learning in the corresponding frequency band. This issue is now directly addressed in the Results section.

Reviewer #2:

*The data and analysis seem solid, but I have significant concerns about the “marketing” of the results. The authors imply in many places that it is somehow more likely that the spike-field coherence is responsible for the learning than a result of the learning. I don't think the authors can or should say more than state the fact that these two features co-occur and are correlated with one another. I don't think it makes the paper more or less interesting if the speculative idea that spike-timing-dependent plasticity is underlying the learning, and that one might expect to see more synchronized activity when spike-timing-dependent plasticity is occurring*.

To improve clarity, we have completely rewritten the second paragraph of the Discussion section, which frames our results in the context of synaptic plasticity and theta oscillations. The reasons why we decided to discuss our findings in this context are the following: (i) The timing between incoming stimuli and ongoing theta oscillations has been found to control synaptic changes, and neuronal oscillations in the theta frequency range (3–8 Hz) have been associated with the induction of synaptic plasticity (see [4], for a review). (ii) Oscillations in the theta range can be modulated by behavior and brain state (40; 13) and have been assigned a key role in the maintenance of information in short-term memory (21; 23). (iii) Our results are consistent with previous evidence in rabbit hippocampus (3) that animals that exhibit more theta activity show elevated classical conditioning of the nictitating membrane response.

*Specific items: The only specific requirements I have are to re-write some of the paper so that it is clear that the authors are only establishing a correlation between these two quantities. The paper reads as though there is some evidence for causation in a particular direction (that the SPC is causing the learning)*.

We agree with the reviewer. Although we never intended to imply causal links between rapid learning and theta spike-field coherence, there were several places in the original manuscript where the writing was confusing, and we might have inadvertently suggested that theta coherence causes learning or vice versa. We have carefully revised the entire manuscript to correct these errors. The examples provided by the reviewer have all been revised to eliminate perceived claims for causal relationships.

Reviewer #3:

*This is a very interesting study linking performance in a perceptual learning task with spike “synchronization” with the LFP in the theta range. The significance is high as it shifts the emphasis from the single unit level to the population level. The data and analysis seems convincing (although I am an expert in this type of quantitative approaches). I have only one general consideration to raise, that the rationale for doing the studies in V4 is not clear. This is important because many forms of perceptual learning do take place in V1, raising the possibility that observation in V4 are secondary to other changes occurring at earlier stages of perception. Perhaps the reason for choosing V4 is obvious, but it needs to be stated*.

We now spell out the rationale for examining rapid learning effects in area V4 as opposed to other areas. To address the reviewer’s criticism we added a new paragraph (the last paragraph in the Introduction section) that describes the reasons for focusing our study on area V4: (i) among all sensory cortical areas, the visual cortex is the best understood in terms of receptive field properties and circuitry, thus it provides a unique opportunity for investigating the impact of perceptual learning on neuronal responses, (ii) area V4, unlike V1, sends inputs to downstream areas involved in perceptual decisions, (iii) individual neuron responses in extrastriate cortex (such as V4) are more strongly correlated to behavior than those in V1, and (iv) previous lesion studies have suggested that area V4 plays a key role in perceptual learning (without demonstrating what that role is).

[Editors' note: further revisions were requested prior to acceptance, as described below.]

*1) Why did learning fail to occur during five additional sessions? What was different about these sessions than the others illustrated? (This should be made clear in the manuscript)*.

We now made clear (please see the Results) that the five control sessions were collected from one animal, and were recorded before the monkey was able to accurately perform the behavioral task with the set of 10 prototype stimuli. That is, these sessions were recorded before the monkey learned the behavioral task – the images served as control stimuli to demonstrate that, as expected, the changes in theta spike-LFP coherence are absent when learning is absent.